# Molecular Mechanisms of Neuroprotection after the Intermittent Exposures of Hypercapnic Hypoxia

**DOI:** 10.3390/ijms25073665

**Published:** 2024-03-25

**Authors:** Pavel P. Tregub, Vladimir P. Kulikov, Irada Ibrahimli, Oksana F. Tregub, Artem V. Volodkin, Michael A. Ignatyuk, Andrey A. Kostin, Dmitrii A. Atiakshin

**Affiliations:** 1Department of Pathophysiology, I.M. Sechenov First Moscow State Medical University, 119991 Moscow, Russia; ibrgmv.i@gmail.com; 2Brain Science Institute, Research Center of Neurology, 125367 Moscow, Russia; 3Scientific and Educational Resource Center “Innovative Technologies of Immunophenotyping, Digital Spatial Profiling and Ultrastructural Analysis”, RUDN University, 117198 Moscow, Russia; volodkin-av@rudn.ru (A.V.V.); ignatyuk-ma@rudn.ru (M.A.I.); andocrey@mail.ru (A.A.K.); atyakshin-da@rudn.ru (D.A.A.); 4Department of Ultrasound and Functional Diagnostics, Altay State Medical University, 656040 Barnaul, Russia; kulikov57@mail.ru; 5Independent Researcher, 127055 Moscow, Russia; tregub.oksana@inbox.ru

**Keywords:** hypoxia, hypercapnia, neuroprotection, blood–brain barrier permeability, apoptosis inhibition, antioxidant systems, chaperones, A1 adenosine receptors, endoplasmic reticulum, mitochondrial ATP-dependent potassium channels, HIF-1α

## Abstract

The review introduces the stages of formation and experimental confirmation of the hypothesis regarding the mutual potentiation of neuroprotective effects of hypoxia and hypercapnia during their combined influence (hypercapnic hypoxia). The main focus is on the mechanisms and signaling pathways involved in the formation of ischemic tolerance in the brain during intermittent hypercapnic hypoxia. Importantly, the combined effect of hypoxia and hypercapnia exerts a more pronounced neuroprotective effect compared to their separate application. Some signaling systems are associated with the predominance of the hypoxic stimulus (HIF-1α, A1 receptors), while others (NF-κB, antioxidant activity, inhibition of apoptosis, maintenance of selective blood–brain barrier permeability) are mainly modulated by hypercapnia. Most of the molecular and cellular mechanisms involved in the formation of brain tolerance to ischemia are due to the contribution of both excess carbon dioxide and oxygen deficiency (ATP-dependent potassium channels, chaperones, endoplasmic reticulum stress, mitochondrial metabolism reprogramming). Overall, experimental studies indicate the dominance of hypercapnia in the neuroprotective effect of its combined action with hypoxia. Recent clinical studies have demonstrated the effectiveness of hypercapnic–hypoxic training in the treatment of childhood cerebral palsy and diabetic polyneuropathy in children. Combining hypercapnic hypoxia with pharmacological modulators of neuro/cardio/cytoprotection signaling pathways is likely to be promising for translating experimental research into clinical medicine.

## 1. Introduction

It is well known that hypoxic exposure is not only a damaging factor but also a method capable of increasing the tolerance of the brain to oxygen deficiency/ischemia [1,2]. To achieve a neuroprotective effect using hypoxic interventions, researchers typically employ preconditioning regimens (FiO_2_ = 8–12%, 1 to 3 sessions of 30–120 min) and intermittent training (FiO_2_ = 9–16%, 5 to 45 sessions of 48–90 min per day), which are conducted 24 h before modeling cerebral ischemia [2]. By “ischemia”, researchers usually mean an episode of prolonged cessation of perfusion (for 10 min or more) in a specific region of the brain, which, in the case of ischemia–reperfusion injury, is accompanied by the restoration of blood flow to the ischemic area [2].

However, there has been a significant increase in interest in studying the therapeutic effectiveness of permissive hypercapnia. Permissive hypercapnia is a widely used ventilation strategy in the practice of intensive care and resuscitation, showing significant protective effects in several models of neuronal damage in vitro and in vivo [3]. It has been demonstrated that carbon dioxide at nontoxic doses (PaCO2 = 60–100 mmHg for 2 h) exerts a protective effect on the brain during ischemia–reperfusion injury [3,4].

In 2009, based on experimental studies showing a pronounced increase in the brain’s tolerance to ischemia under the influence of hypercapnic–hypoxic training, the hypothesis of potentiation of the neuroprotective effects of hypoxia and hypercapnia under their combined action was formulated [5]. In 2013, experimental confirmation of this hypothesis was obtained in a comparative study of the isolated and combined effects of hypoxia and hypercapnia [6]. In the same year, the hypothesis was supplemented by the concept of the dominance of hypercapnia in the effect of forming the body’s resistance to acute oxygen deficiency under its combined action with hypoxia [7]. Subsequently, on a model of ischemic stroke in rats, it was shown that the combined effect of hypoxia and hypercapnia (hypercapnic hypoxia) significantly reduces the volume of brain damage compared to their isolated effects [8].

However, the neuroprotective mechanisms underlying this effectiveness have been poorly studied. At the same time, these data provide a fundamental basis for the development of clinical methods of respiratory training, including in combination with pharmacological agents, for the prevention and treatment of stroke, vascular dementia, neurodegenerative diseases, and perinatal hypoxic injuries.

Among the basic molecular and cellular mechanisms of the formation of ischemic tolerance of the brain are the maintenance of electrolyte balance and mitochondrial metabolism of nerve cells, adaptive effects of mitoK^+^_ATP_-channels and adenosine receptors, protection of cells from free radicals, mobilization of endoplasmic reticulum, effects of neurotrophic factors and chaperones, neuro- and synaptogenesis, inhibition of apoptosis, and preservation of selective permeability of the blood–brain barrier, as well as epigenetic effects of HIF-1α [9,10,11,12].

This review aims to systematize the available information on neuroprotective mechanisms that form ischemic tolerance of the brain after intermittent exposure to hypercapnic hypoxia and to determine potential correlations between different signaling pathways of these mechanisms. These data complement our hypothesis about the effect of mutual potentiation of the neuroprotective effects of hypoxia and hypercapnia under their combined action and the important role of the hypercapnic component in this mechanism.

## 2. Functioning of Intracellular Systems, Organelles, and Messengers

### 2.1. Maintenance of Electrolyte Balance and Ca^2+^ Homeostasis

The distribution of electrolytes between the intra- and extracellular environments, as well as the levels of Ca^2+^ ions within the cytoplasm and cellular organelles, determine the viability of neurons in the ischemia-tolerant brain [13]. Therefore, many neuroprotective mechanisms actively modulate signaling pathways involved in regulating this balance. For example, it is known that in the cell membrane of ischemia-resistant neurons in the CA1 region of the hippocampus, there is high activity of Ca^2+^-ATPase and enhanced binding of Ca^2+^ in mitochondria [14,15]. The intracellular calcium level in neurons of this hippocampal area in animals tolerant to ischemia is significantly reduced after an episode of anoxia–aglycemia [16]. It has also been shown that after preconditioning with 3-nitropropionic acid, the expression of Ca^2+^-ATPase is enhanced in the cytoplasmic membrane of hippocampal neurons [17]. Regarding the stabilizing effect of hypercapnia on intracellular calcium levels, some conclusions can be extrapolated from data on the process of pH/P_CO2_ chemoreception by the carotid body [18] and mast cell degranulation [19].

It is important to note that modeling transient global ischemia in rats enhances the expression of Na^+^/Ca^2+^ exchanger genes in neurons, along with the expression of heat shock proteins [20]. In this context, the main role in neurodamage is predominantly played by the sodium–calcium exchanger genes NCX1 and NCX3, which undergo increased expression during brain ischemia [21], while upregulation of NCX2 expression participates in the mechanism of neuroprotection induced by ischemic preconditioning [22].

The Na^+^/H^+^ exchanger and the Na^+^/K^+^/2Cl^−^ cotransporter are also important ion transporters as they participate in the regulation of acid–base balance and cellular fluid volume [23]. In this regard, data on the inhibition of Na^+^/H^+^ exchanger (NHE1) expression in pyramidal neurons of the CA1 region of the hippocampus after preconditioning episodes are important [24]. Evidence has also been obtained confirming the involvement of the interaction between the endoplasmic reticulum and mitochondria in the preconditioning mechanism [11]. In particular, the key role of NCX isoforms in regulating Ca^2+^ homeostasis in different subcellular compartments has been discussed in establishing an ischemia-tolerant phenotype in neurons.

Recently, it was established that in hippocampal neurons from the CA1 region, after exposure to permissive hypercapnia, the size of mitochondria increased, and the membrane compartments of the endoplasmic reticulum expanded, while after exposure to normobaric hypoxia, only the size of mitochondria increased [25]. Based on these data, it can be assumed that these membrane structures, which support electrolyte and calcium homeostasis, may be involved in the process of ischemia tolerance formation under the combined influence of hypercapnia and hypoxia. However, detailed information for the central nervous system has not yet been obtained.

Most of the information about the mechanisms of neuron resilience to ischemia/hypoxia mentioned in this section has been studied regarding neurons in the CA1 region of the hippocampus. This is because the CA1 region exhibits the highest sensitivity to oxygen deficiency; therefore, the increase in neuron resilience to ischemia/hypoxia in this area of the brain is considered a characteristic feature of the formation of ischemic tolerance in the brain. Data on neuroprotective mechanisms can be extrapolated to other areas of the brain [9].

### 2.2. Reprogramming of Cellular Metabolism

One of the earliest and most effective mechanisms for the formation of ischemic tolerance in the brain involves signaling pathways that optimize cellular energetics under conditions of oxygen–glucose deprivation [26]. Most of the information in this context pertains to the adaptogenic effects of hypoxia, which has a pronounced influence on the mitochondrial metabolism of nerve cells [27,28]. Regarding energy systems, it is known that the neuroprotective effects of intermittent hypoxia are accompanied by a decrease in the content of respiratory carriers at the terminal segment of the respiratory chain and a reduction in their oxidative capacity [29]. Additionally, mitochondrial enzymatic complexes undergo restructuring to operate in a more efficient mode, achieved by increasing the efficiency of oxidative phosphorylation and the number of mitochondria in the cell [30].

The utilization of energy substrates after adaptogenic hypoxic exposure also undergoes modifications. For instance, in an experiment by Brucklacher R.M. et al. [31], interesting results were obtained: the level of glycogen in the rat brain increased 24 h after hypoxic preconditioning, and the level of residual ATP at the end of a 90 min hypoxia–ischemia was significantly higher than in the control group. Additionally, it was recently shown that the increased intracellular level of lactate has a neuroprotective effect [32]. These data may describe a complex of adaptive metabolic changes in the brain during the formation of ischemic tolerance after intermittent hypoxic exposure.

It is known that glucose uptake and its utilization in brain tissues primarily occur with the involvement of glucose transporter proteins GLUT1 and GLUT3, and their modulation can increase ischemic tolerance [33]. It has also been shown that the expression of GLUT1 is enhanced by the action of the HIF-1α factor, which accumulates under hypoxic conditions [34], and the antiapoptotic factor bcl-2 [35], activated by permissive hypercapnia [3].

It is important to consider the acidogenic potential of carbon dioxide, which reduces both the activity of oxidative phosphorylation and the production of reactive oxygen species in mitochondria [36]. These effects, occurring under conditions of permissive hypercapnia, likely influence the signaling pathways optimizing cellular metabolism in the mechanism of neuroprotection. This is supported by data on the influence of permissive hypercapnia on increasing the sizes of mitochondria in hippocampal neurons [25].

### 2.3. Remodeling of Intracellular Signaling Associated with Adenosine, Its Receptors, and mitoK^+^_ATP_ Channels

Adenosine plays a significant role in increasing the tolerance of the brain to ischemia when exposed to hypoxia [37]. It acts on the corresponding A1 receptors, which activate protein kinase C [37]. This signaling cascade leads to modification of mitoK^+^_ATP_ channels, which decreases ATP synaptic transmission, shifting the CNS balance towards inhibition [38]. These signaling molecules are of great importance both in the protective mechanism of preconditioning and in the formation of stable tolerance of the brain to ischemia [39,40].

Hypoxic preconditioning exerts effects on adenosine and A1 receptors similar to ischemic preconditioning [41]. For example, it has been shown that exposure to four cycles of hypoxia in a preconditioning regimen in mice maintains the density of A1 receptors and increases their affinity to adenosine in the CA1 region of the hippocampus, brainstem, and medulla oblongata, leading to a neuroprotective effect [42], while blockade of cyclopentyladenosine A1 receptors abolishes the protective effect of both adenosine administration and hypoxic preconditioning during global cerebral ischemia [43]. These findings are supported by experiments with anoxic exposure on isolated hippocampal slices under preconditioning conditions [44]. It has also been established that reducing the level of extracellular adenosine leads to the loss of hypoxia-induced neuroprotection after intermittent hypoxic exposure, which is directly associated with HIF-regulated expression of target genes [45]. Additionally, the authors of [46] demonstrated that activation of A1 receptors reduces the chemosensitivity of respiratory neurons to increased CO_2_ levels but not to decreased O_2_ levels. This suggests a possible biological antagonism of hypercapnia and hypoxia when acting on these receptors.

Mitochondrial ATP-sensitive potassium channels are considered the final effector of intracellular signaling after preconditioning [47,48], and their activation is an important mechanism of neuroprotection [49]. Particularly significant in this regard are the following studies: Zhang S. et al. [50], which demonstrated an increase in the expression and activity of SUR1 and Kir6.2 (subunits of mitoK^+^_ATP_ channels) in samples of ischemic rat brain after prolonged intermittent hypoxia, which were abolished by the administration of the 5-hydroxydecanoate channel blocker; Sun H.S. et al. [51], which found that exposure to hypoxia prior to hypoxic–ischemic injury induced an elevation in the level of Kir6.2 isoform protein and enhanced the current activity of mitoKATP channels. Additionally, an interesting observation is that hypercapnia leads to the activation of calcium-activated and ATP-dependent potassium channels [52].

The evaluation of the impact of hypercapnia and hypoxia on the intracellular signaling molecules described above, conducted on in vitro cellular models, reveals several important findings [53]. Specifically, a stimulating effect of intermittent hypoxia, but not hypercapnia, on the relative content of cells with A1 receptors in astrocytes after their isolation in culture was discovered. This indicates the absence of a direct influence of CO_2_ on these receptors when combined with hypoxic exposure. Additionally, it demonstrates that intermittent hypoxia influences the epigenetic regulation of A1 receptors’ levels in astrocytes even after their cultivation. This could be explained by the effects of the transcription factor HIF-1α, which enhances the activity of adenosine receptors [54], or by the influence of hypoxia on G proteins coupled to these receptors [55].

It has also been demonstrated that the assessment of the percentage of cells with A1 receptors in astrocyte cultures after chemical hypoxia in vitro shows a protective effect of combined hypercapnia and hypoxia exposure [53]. This effect was observed both in astrocyte cultures isolated from rats exposed to hypercapnic hypoxia in vivo and in astrocyte cultures subjected to equivalent exposure in vitro. It is reasonable to assume that the results obtained regarding A1 receptors, which are less characteristic for astrocytes, will also be applicable to neurons [56].

Under conditions of hypoxia and ischemia, there is an increase in the concentration of reactive oxygen species [57] and changes in the ratios of GSH/GSSG and NAD^+^/NADH, leading to the modification of cysteine thiol groups in membrane structures [9]. Moreover, the elevation of intracellular levels of free oxygen radicals leads to pronounced activation of mitoK^+^_ATP_ channels [58,59]. This is further facilitated by the enhancement of NO° synthesis with the formation of peroxynitrite [60] and subsequent activation of protein kinase C [61]. Additionally, there is evidence that the selective activation of mitochondrial ATP-dependent K^+^ channels in astrocytes induces delayed preconditioning [62] and increases glutamate uptake in culture, which may provide additional protective benefits [63].

The assessment of the percentage of astrocytes expressing mitoK^+^_ATP_ channels in vitro, cultured after exposure to normobaric hypoxia and permissive hypercapnia in vivo, indicates the possibility of their regulation through epigenetic mechanisms. This could be attributed to O_2_-deficient oxidative stress [64] and the activation of protein kinase C induced by increased CO_2_/bicarbonate concentration and intracellular Ca^2+^ elevation [9]. Additionally, the impact on astrocytes by sodium iodoacetate demonstrated a stabilizing effect of hypercapnia on the regulation of the percentage of cells containing mitoK^+^_ATP_ channels.

Modeling focal (local) ischemic brain damage in rats using the photothrombosis method of microvessels in the area of the temporal cortex, after exposure to hypercapnia and/or hypoxia, revealed trends similar to the results obtained in vitro [65]. It was found that permissive hypercapnia reduces the percentage of cells with A1 receptors in the peri-infarct area of the rat brain and also decreases it when combined with normobaric hypoxia. At the same time, permissive hypercapnia increases the relative content of cells with mitoK^+^_ATP_ channels both with isolated exposure and in combination with normobaric hypoxia (Figure 1) [53].

These findings are supported by the fact that administering blockers of A1 receptors and mitoK^+^_ATP_ channels to rats prior to the combined exposure of hypercapnia and hypoxia prevents the development of resistance to acute oxygen deficiency [65]. Interestingly, blockade of adenosine receptors did not affect the protective efficacy of permissive hypercapnia, unlike normobaric hypoxia. Meanwhile, blockade of mitoK^+^_ATP_ channels prevented the development of resistance to acute hypoxia in both permissive hypercapnia and normobaric hypoxia.

### 2.4. Activation of Antioxidant Systems

There is ample evidence that free radical mechanisms play a role in the process of brain damage caused by ischemia/reperfusion [66,67]. During reperfusion, there is an excessive release of nitrous oxide, which can also contribute to ischemic tissue damage by generating reactive nitrogen species such as peroxynitrite (product of the reaction of nitric oxide with superoxide radicals) [68]. Due to its high penetrating ability, peroxynitrite penetrates the plasma membrane more easily than superoxide anion, oxidizing intracellular molecules. This leads to tyrosine protein nitration, lipid peroxidation, mitochondrial dysfunction, and DNA damage [68].

Data suggest that the generation of the superoxide anion during preconditioning is necessary for the subsequent development of ischemic tolerance [69,70]. For instance, intravenous administration of recombinant superoxide dismutase to rats before ischemic preconditioning prevented the development of tolerance to subsequent transient ischemia, similar to enhancing the expression of HSP-70 [71]. Similar data exist regarding the protective properties of NO°, involving the induction of gene expression underlying brain preconditioning [72,73].

It is noteworthy that moderate concentrations of CO_2_ stimulate antioxidant activity [74,75,76]. Specifically, this mechanism is mediated by the activation of glutathione peroxidase [74] and superoxide dismutase [75,76]. Moreover, carbon dioxide may influence the stability of the iron-transferrin complex [76], preventing iron ions from participating in initiating free radical reactions [77]. It is also well known that the CO_2_ molecule can neutralize reactive oxygen/nitrogen species by reacting with peroxynitrite and converting to nitrocarbonate, which forms a carboxyl anion and a nitroxide anion upon reaction with water [78,79] (Figure 2). Notably, in their experiment, the authors of [80] demonstrated that moderate hypoxia and hypercapnia contribute to the activation of the antioxidant system in vivo damage by increasing the expression of genes encoding cytoplasmic superoxide dismutase, glutathione peroxidase, and peptide methionine sulfoxide reductase.

### 2.5. Adaptive Endoplasmic Reticulum Response and Chaperone Cascade

Increased expression of heat shock protein genes is a universal cellular response to damage, and their chaperone activity provides cytoprotection during stress [81]. In vivo studies have shown that tolerance to cerebral ischemia is directly related to the induction of nonconstitutive HSP-70 (or GRP-78) and other chaperones [82], and early in vitro studies have already confirmed that HSP-70 inactivation weakens adaptive cyto- and neuroprotection [83] The importance of HSP-70 in implementing the mechanism of adaptive cytoprotection and the neuroprotective properties of its high levels are convincingly argued [84], as well as its role in immune suppression induced by ischemic stroke [85]. The main functions of heat shock proteins, particularly HSP-70 (or GRP-78), include protein folding during ribosomal synthesis, prevention of refolding of damaged proteins during intracellular stress, and transmembrane protein transport [9]. It is also known to undergo phosphorylation, which regulates some of its functions [81].

It is also known that HSP-70 can act as a direct antagonist of apoptosis [81]. HSP-70-induced mitochondrial protection suggests stimulation of cell survival after ischemic injury, and such an antiapoptotic effect is supported by the fact that HSP-70 overexpression reduced the release of cytochrome C from mitochondria into the intermembrane space [86] and nuclear translocation of apoptosis-inducing factor (AIF) [87].

Significant interest lies in the adaptive endoplasmic reticulum response during stress in nerve cells, regulated by the chaperone cascade [88,89]. Key signaling mechanisms of this cascade include the aforementioned chaperone GRP-78 (or HSP-70) [88] and NF-κB, considered one of the main signaling “transmitters” of preconditioning [90]. Additionally, there is evidence that HSP-70 can regulate NF-κB activation [91], mediating its antiapoptotic function through the activation of the PKR/NF-κB-dependent protective pathway [92].

It has been found that the expression of the chaperone GRP-78 is lowest after exposure to normobaric hypoxia [93], indicating pronounced disturbances in cellular metabolism and deactivation of the adaptive potential of neurons, leading to ER dysfunction. This fact may be due to the depletion of adaptogenic mechanisms in the ER after a 15-course exposure to hypoxic conditions. Meanwhile, the content of GRP-78 after exposure to hypercapnic hypoxia and permissive hypercapnia was higher than after exposure to hypoxia. This may suggest that hypercapnia in combination with hypoxia has a pronounced effect on the activation of the chaperone GRP-78, with carbon dioxide having a dominant influence in this process (Figure 3). A similar trend was observed in the study of HSP-70 in the serum of rats [94].

The transcription factor NF-κB, the level of which was assessed simultaneously with GRP-78 [93], is typically found in an inactive state in the cytosol under resting conditions. During stress and activation, it translocates into the nucleus, activating the transcription of genes that inhibit apoptosis and promote cellular adaptation. It is also a key regulator of initiating the tissue’s proinflammatory response to stress [95].

After isolated exposures to hypoxia and hypercapnia, NF-κB expression in the peri-infarct area was also found to be more intense than in controls [93]. This could be attributed to both the similar activity of hypercapnia and hypoxia in stimulating this factor and the translocation of the active form of NF-κB from the cytoplasm to the nucleus. Data showing that moderate intermittent hypoxia activates NF-κB in the hippocampus and neocortex, thereby contributing to the development of tolerance to ischemia/hypoxia [96], supporting the results obtained regarding normobaric hypoxia.

The content of NF-κB in the cytoplasm of nerve cells was highest after the combined exposure to hypercapnia and hypoxia. The level of nuclear expression of NF-κB was also most pronounced with the combined exposure to hypercapnia and hypoxia. In the group exposed to permissive hypercapnia, the expression was less intense but significant compared to the control level and normobaric hypoxia. This may suggest that the translocation of this factor into the nucleus did not occur during hypoxia, whereas during hypercapnia, there was an increase in cytoplasmic expression of NF-κB in combination with an increase in its nuclear concentration, indicating the translocation of the activated factor into the nucleus. Taking this into account, the highest cytoplasmic and nuclear expression of NF-κB in the hypercapnic hypoxia group can be explained by the fact that the combined effects lead to a summation of effects, subsequently resulting in more pronounced neuroprotection. On the other hand, hypoxia and hypercapnia may activate different pathways for NF-κB activation.

Thus, regarding the activation of the adaptive branch of the ER, it can be assumed that the combined exposure to hypoxia and hypercapnia maximally increases the expression of the chaperone GRP-78 and the NF-κB factor. In this context, the hypercapnic component, when combined with hypoxia, dominates in the signaling pathway for the activation of GRP-78 and the transcription factor NF-κB.

## 3. Modulation of the Life Cycle of Nerve Cells

### 3.1. Proliferative and Synthetic Activity

The potential proliferation of precursor stem cells provides the mature brain with flexibility and self-renewal through neurogenesis, which occurs in response to external stimuli and damage [97,98]. It has been shown that in the ischemia-tolerant brain, the proliferation of precursor cells increases after focal ischemia and preconditioning [99,100]. Meanwhile, synaptogenesis is another potential mechanism of variability and self-renewal after ischemic brain damage [101,102].

It is known that normobaric hypoxia activates synthetic and proliferative activity in cells [103], and a number of recent experimental studies have shown that hypoxic preconditioning enhances the proliferative activity of mesenchymal stem cells [104,105], as well as neural stem cells [106].

The results obtained from assessing the morphogenetic parameters of nerve cells following hypercapnic–hypoxic preconditioning prior to focal ischemic brain injury showed the dominant influence of CO_2_ on the activation of synthetic activity in nerve cells [107]. This was manifested by an increase in the number of nucleoli in nerve cells, resulting from increased synthesis of ribosomal RNA on nucleolus-forming chromosomes [108], leading to enhanced synthesis of neurotransmitters in neurons, structural proteins, and neurotrophic factors in glial cells [109]. This process plays an important role in the mechanism of neuroprotection against ischemic damage to nervous tissue, as the increased synthetic function in microglial cells helps reduce the consequences of trophic disturbances, and the elevated expression of neurotransmitters holds high reparative value for neurons in the phase of paranecrosis and necrobiosis [9].

Monitoring the cellular index (a measure of proliferative activity in the xCELLigence^®^ RTCA system) in astrocyte–neuron cocultures in vitro after combined and isolated exposure to hypercapnia and/or hypoxia demonstrated the following facts [110]:Oxygen–glucose deprivation only suppresses the activity of intact nerve cells for the first 12 h of observation.Intermittent hypoxic exposure to astrocyte–neuron cocultures stimulates cellular activity, which remains elevated throughout the observation period in vitro.Acute oxygen–glucose deprivation does not affect the dynamics of the cellular index in cocultures of cells obtained from rats after exposure to normobaric hypoxia in vivo, indicating its protective effectiveness.Permissive hypercapnia delayedly increases cellular proliferative activity with the formation of a prolonged latent period, which also indicates the protective potential of CO_2_.

For hypoxic exposure, these effects may be due to the stimulating influence of intermittent hypoxia on the functioning of signaling pathways activating the transcription factor HIF-1α, which triggers antiapoptotic mechanisms in cells [111,112], often associated with many oncogenic hyperproliferative states [113]. In turn, permissive hypercapnia likely enhances the proliferative activity of nerve cells due to the inhibitory effect of carbon dioxide on apoptosis in nerve cells [3], as well as the activation of MAPK and PI3K systems in microglia, leading to increased synthesis of HIF-1α via the oxygen-independent pathway [111,114,115]. Additionally, the signaling pathways of antiapoptotic mechanisms during hypoxia and hypercapnia exposure will be detailed in the following section.

Thus, it can be asserted that the mechanism of neuroprotection during combined hypoxia and hypercapnia exposure plays an important role in increasing cellular activity and synthetic function in nerve cells, with the hypercapnic component being of paramount importance in it (Figure 4).

### 3.2. Modulation of Apoptosis

One of the most crucial mechanisms enhancing the brain’s tolerance to ischemia is the inhibition of apoptosis in the peri-infarct zone [9,116]. This protective mechanism during the reperfusion period prevents the death of partially damaged neurons in the ischemic zone. Cells in the infarct core typically die through necrosis, while cells in the peri-infarct zone undergo apoptosis [117]. It is well known that cell death due to necrosis results in the loss of cellular membrane integrity and uncontrolled release of cell death products into the extracellular space, initiating an inflammatory response in surrounding tissues. On the other hand, cell death due to apoptosis allows for the avoidance of this, as it concludes with the formation of apoptotic bodies (enclosed by plasma membrane) [9]. The primary factor determining the cell death mechanism is the level of ATP inside the cell [118]. Additionally, in the context of therapeutic intervention, apoptosis is preferable to necrosis, as it can be blocked by various treatment methods, allowing the preservation of partially damaged tissue [116,119]. In other words, the more damaged cells in the ischemic injury zone that undergo apoptosis (rather than necrosis), the greater the chances of preserving their viability (including through pharmacological neuroprotectants) and reducing the level of inflammatory reaction in the stroke focus.

There is ample evidence that hypoxic exposure (both in preconditioning and intermittent influence) can exert an antiapoptotic effect. For instance, a study by Cantagrel S. et al. [120] reported a reduction in apoptotic cells in the brain preconditioned with hypoxic exposure 24 and 48 h after experimental stroke. In the work of Coimbra-Costa D. et al. [121], it was found that eight episodes of 3 h intermittent hypobaric hypoxia exposure in rats led to a decrease in apoptotic protein levels in the brain astrocytes during acute damaging hypoxia. Additionally, there is evidence of activation of the transcription factor HIF-1α in cortical neurons following moderate hypoxic exposure, which led to the inactivation of the p53 protein (which halts the cell cycle with DNA replication and initiates apoptosis) and slowed down delayed cell death [122]. It should also be noted that a 6 h exposure to severe hypoxia (FA_O2_ = 7%), including the overexpression of HIF-1α and oxidative stress, led to increased activity of cytochrome C, AIF, and caspase-3 in the hippocampus during a 24 h reoxygenation period [123]. The apoptosis-inhibiting effect of normobaric hypoxia, as mentioned earlier, may be associated with the activation of the PI3K system and the antiapoptotic effects of HIF-1α [111,114,115].

It is known that the chaperone HSP-70, whose activity increases after hypoxic preconditioning, inhibits apoptosis by stimulating the PKR/NF-κB-dependent pathway [92]. Additionally, HSP-70 inhibits reactions that promote increased permeability of mitochondrial membranes and the release of cytochrome C by blocking Bax and increasing the expression of the antiapoptotic factor Bcl-2 [124]. An interesting fact is that specific domains of HSP-70 are involved in preventing the release of mitochondrial AIF and sequestering AIF in the cytosol, supporting the hypothesis that HSP-70 has the potential to suppress cell death through various mechanisms [125]. Activation of mitoK^+^_ATP_ channels, induced by hypercapnic and hypoxic exposures, prevents the activation of cytochrome C and, consequently, also blocks the caspase-dependent apoptotic pathway [126].

In recent years, researchers from China have extensively elucidated the mechanisms of apoptosis inhibition during the reperfusion period following transient cerebral ischemia by permissive hypercapnia [3]. The authors demonstrated that inhalation of CO_2_ at a moderate concentration (Pa_CO2_ = 60–100 mmHg) inhibits the active form of the major effector caspase-3, reduces the cytosolic content of cytochrome C and proapoptotic protein Bax, and increases the concentration of the antiapoptotic protein Bcl-2 in mitochondria. Moreover, based on the evaluation of apoptosis in the peri-infarct area during focal ischemia preceded by intermittent hypercapnia and/or hypoxia exposures, it was shown that permissive hypercapnia, including in combination with hypoxia, exerts an inhibitory effect on apoptosis [127].

In another experimental study, the content of cells with apoptotic inducers in the peri-infarct area of the rat brain was investigated after exposure to hypercapnia and/or hypoxia (caspase-3, AIF, Bax, Bcl-2) [128]. The content of cells with proapoptotic factors (caspase-3, AIF, and Bax) in the cells of the peri-infarct area of the brain decreased after exposure to permissive hypercapnia and hypercapnic hypoxia, while the number of cells with the antiapoptotic factor Bcl-2 increased after all modes of both combined and isolated exposure to hypercapnia and hypoxia. Such effects of hypercapnia, primarily acting on the mitochondrial apoptotic pathway, may be due to its antioxidant effects (see Section 2.4), as well as the stabilization of the NAD^+^/NADH ratio and the buffering effect of bicarbonate on Ca^2+^ [129].

The assessment of the relative content of cells with proapoptotic (caspase-3, Bax, and AIF) and antiapoptotic (bcl-2) factors in astrocytes and neurons in vitro after hypercapnic and/or hypoxic exposure preceding oxygen–glucose deprivation demonstrated comparable results [128]. A pronounced effect of inhibiting apoptotic signaling pathways was also observed after exposure regimes combining permissive hypercapnia and normobaric hypoxia, and when they were intermittently applied, a positive effect was observed in both astrocytes and neurons, while with continuous exposure, it was observed only in neuronal cells. This may suggest that intermittent modes of moderate hypercapnia and hypoxia exert a more pronounced neuroprotective effect, affecting not only neurons but also glial cells, thus achieving high tolerance to ischemia [3,130].

The available experimental data on in vivo/in vitro models suggest that hypercapnic exposure, both alone and in combination with hypoxia, leads to a reduction in the content of cells with proapoptotic mediators and an increase in cells with antiapoptotic mediators (Figure 5) [128]. This can be considered part of the mechanism of inhibiting apoptosis and increasing the tolerance of the brain to damaging hypoxia/ischemia. The hypoxic component, when combined with hypercapnia, also contributes to this neuroprotective mechanism, mainly through indirect pathways.

## 4. Maintenance of Selective Permeability of the Blood–Brain Barrier

Researchers emphasize the role of blood–brain barrier (BBB) dysfunction in the development of various neurological disorders [131]. Impairment of BBB integrity and permeability is a crucial element in the pathogenesis of hypoxic/ischemic and infectious brain injury [132]. A multilevel system of chemical regulation of tissue homeostasis in the brain has been identified, enhancing neuronal protection against stress and damage, as well as the role of numerous growth and neurotrophic factors in reparative processes, largely determining the outcomes and prognosis of brain hypoxic injury [133].

Several studies have shown that ischemic tolerance contributes to the preservation of the BBB and reduces edema formation during controlled ischemia [134,135]. From a mechanistic standpoint, it is clear that reducing inflammatory reactivity in the brain likely contributes to preserving BBB integrity [136]. At the same time, preserving BBB integrity may be an important feature of ischemic tolerance, considering the interaction between damaged/vulnerable tissue and the circulation of inflammatory cells during reperfusion.

Evaluation of BBB permeability following exposure to hypercapnia and/or hypoxia showed a stimulating effect of the hypercapnic component on maintaining the concentration of Evans blue dye in the blood plasma at an elevated level even 24 h after its intraperitoneal administration, while the indicators of the normobaric hypoxia group were virtually indistinguishable from the control values [137]. In the same study, the optical density of the dye measured in the brain tissue was also elevated in rats from the hypercapnic hypoxia group. These findings are likely associated with stimulation of angiogenesis during hypercapnic–hypoxic training [94], as well as with the influence of CO_2_ and acidosis on the constriction of resistive arterioles in peripheral organs and the dilation of arterioles in the myocardium and brain [138]. Against this background, the moderate increase in dye content in brain tissue in the hypercapnic hypoxia group is more objectively considered in the context of the BBB permeability index, which takes into account the dye content in both blood plasma and the brain.

Considering that the combined effect of intermittent hypercapnia and hypoxia resulted in the lowest level of BBB permeability, it can be assumed that both of these components influence this neuroprotective mechanism by activating different signaling pathways.

These results suggest that reduced BBB permeability is one of the probable mechanisms underlying the formation of brain tolerance to ischemia after intermittent hypercapnic exposure (including in combination with hypoxia). Supporting the neuroprotective effectiveness of this mechanism are also the results demonstrating the positive effect of 3 h permissive hypercapnia (60 and 80 mmHg) on BBB integrity when combined with hypoxia in conditions of experimental ischemic brain injury [139]. As shown by the authors, this was due to a decrease in the expression of the AQP4 protein, resulting in reduced brain edema.

It is known that matrix metalloproteinases, particularly MMP-9, disrupt the neurovascular matrix during reperfusion, thereby leading to blood–brain barrier breakdown [140]. It has been shown that BBB integrity disruption, along with MMP-9 expression, was reduced after experimental ischemia in rats preceded by ischemic preconditioning [141]. This is associated with the effects of HIF-1α, which induces the expression of the chaperone HSP-70, which in turn suppresses MMP-9.

Many events in intercellular interactions at the BBB level are determined by the activity of the transcription factor HIF-1, which mediates the cellular response to hypoxia [142]. It is known that HIF-1-induced reactions in energy metabolism are reflected in changes in glycolysis processes, lactate accumulation, and alterations in the nature of neuron–astroglial metabolic coupling [143]. Among the genes controlled by HIF-1 are those encoding stromal cell-derived factor 1 (SDF-1), glycolysis enzymes, and glucose and lactate transporters, which are necessary for cell functioning under conditions of acute or chronic hypoxia [144].

Adhesion molecules released by endothelial cells (e.g., ICAM-1) mediate the firm adhesion of leukocytes to the vessel lining and also trigger signaling cascades that contribute to increased BBB permeability and leukocyte infiltration [145]. Increased ICAM-1 expression by cerebral endothelial cells has been observed during ischemia/reperfusion [146]. Hypoxic preconditioning, on the other hand, reduces the inflammatory response and blocks elevated levels of ICAM-1, as well as inhibits neutrophil adhesion to endothelial cells [147].

An important role in the metabolic regulation of cerebral endothelial activity during ischemic injury is attributed to lactate production, intercellular lactate transport, and utilization [148]. Lactate production is closely associated with the cellular redox state, particularly the ratio of NAD^+^/NADH in mitochondria [149], and this ratio is actively influenced by the effects of hypoxia and hypercapnia, as discussed in Section 2.4.

Recent studies demonstrate the significance of epigenetic regulation in maintaining the selective permeability and integrity of the BBB, which exert a neuroprotective effect under the influence of microRNAs (miRNAs) [150]. For instance, the overexpression of miRNA-126-3p/-5p in the ischemic brains of mice suppresses the effects of proinflammatory cytokines and adhesion molecules, preserving the integrity of cerebral vessel endothelium and reducing the negative consequences of stroke [151]. In turn, this molecule is activated by HIF-1α induced by hypoxic exposure [152]. Conversely, miRNA-34a is activated in endothelial cells during episodes of acute hypoxia/ischemia, negatively affecting mitochondrial function in these cells by impacting cytochrome C [153], while hypoxic preconditioning inhibits this molecule [154].

Thus, the structure of the BBB under conditions of hypoxic/ischemic damage is under the regulatory influence of a range of molecular-cellular systems [155]: epigenetic regulation of cellular metabolism and neuroinflammation by transcription factors; trophic and membrane factors supporting structural integrity and limited permeability; neuron–astroglial metabolic coupling. Moreover, the combined effects of hypercapnia and hypoxia can influence all these systems to varying degrees, forming a neuroprotective effect (Figure 6).

It should be noted that there are also data regarding the negative impact of hypoxic preconditioning on the integrity of the BBB. For instance, Chi O.Z. et al. demonstrated that hypoxic preconditioning exacerbates BBB disruption through the VEGF signaling pathway, suggesting the possibility of worsening brain edema during cerebral ischemia [156]. However, the authors in the mentioned study focused on the short-term (1 h) vasogenic effects of hypoxic preconditioning and their influence on BBB permeability, equating them with characteristics of its integrity.

## 5. Effects of the Transcription Factor HIF-1α

In recent decades, there has been increasing interest in studying the biological effects of the molecule hypoxia-inducible factor 1-alpha (HIF-1α). Such attention is primarily due to its key role in the mechanism of cellular and tissue adaptation to oxygen deficiency and ischemia, as evidenced by a number of works by authoritative scientists [30,157,158].

The HIF-1 factor plays an important role in the cellular response to changes in oxygen homeostasis in mammals, and the main function of this protein is the induction of gene transcription, regulating cellular oxygenation and increasing their tolerance to hypoxia/ischemia [157,158]. The number of discovered target genes activated by HIF-1 continues to increase and includes genes involved in angiogenesis, energy metabolism, erythropoiesis, cell proliferation, vascular remodeling, and vasomotor reactions [159,160]. Moreover, activated HIF-1 exhibits proinflammatory and antimicrobial effects by modulating the cellular immune response [161] and shows proapoptotic effects specific to certain cell types [162].

However, despite traditional views on the nature of the HIF-1α signaling mechanism, based on oxygen-deficit accumulation, information has emerged about alternative (noncanonical) mechanisms of its activation [163]. Such data are of high interest for reconsidering the relationship with this transcription factor, which is regulated not only by hypoxic stimuli but may also serve as a target for potentiating protective effects from several adaptogenic triggers. For example, the synthesis of HIF-1α can be realized through oxygen-independent mechanisms, via reactions controlled by MAPK and PI3K systems, which are important in growth, proliferation, and differentiation processes [111,114,115]. Additionally, it is known that increased transcriptional activity of HIF-1 is observed under the influence of nitric oxide, TNFα, IL-1, and angiotensin [164]. Furthermore, there is information about the degradation mechanism of HIF-1 through the chaperone-mediated lysosomal autophagy pathway [165,166].

It should be noted that in many experimental studies on the classical O_2_-dependent mechanism of HIF-1 activation, models are used that induce not only tissue oxygen deficiency but also a concomitant increase in CO_2_ levels. Carbon dioxide, in turn, is also a significant factor influencing intracellular homeostasis and may exert oxygen-independent effects on HIF-1 activity.

There are studies demonstrating the stimulating effect of hypercapnia on HIF-1α. For instance, in 2009, it was found that the administration of acetazolamide led to an increase in HIF-1α concentration in the brain [167]. The authors positioned this as a direct effect of the administered agent, although the main mechanism of its action involves carbonic anhydrase inhibition with subsequent CO_2_ level elevation, which hypothetically could also be the cause of the observed effect. In another study, Benderro GF et al. [168] assessed the levels of HIF-1α and HIF-2α in the brain after 3 weeks of exposure to chronic hyperoxia and hypercapnia, noting their increased accumulation.

It is important to note that there are publications with opposite conclusions regarding the influence of hypercapnia on HIF-1α levels. For example, in an experimental study by Selfridge A.C et al. [169], hypercapnia suppressed the stability of HIF protein and the expression of its target gene in vivo and in vitro, which was associated with a direct decrease in intracellular pH. Additionally, in the study by Raeis V.B. et al. [170], no correlation was found between increased HIF-1α expression and CO_2_ levels after 24 h of exposure of cardiomyocytes to hypercapnic hypoxia conditions in vitro. However, in these studies, HIF protein suppression was observed when using a model with a high level of hypercapnia (Fet_CO2_ = 10%) and/or prolonged exposure (6–24 h), which could induce a maladaptive effect.

When studying the role of HIF-1α under conditions combining hypoxia with hypercapnia, it was found that permissive hypercapnia did not induce any direct changes in HIF-1α levels, while hypercapnic hypoxia led to increased HIF-1α content in the hippocampal tissues, astrocytes in vitro, and peri-infarct region of rat brains [171]. After chemical hypoxia, the content of HIF-1α in astrocytes in vitro was higher following exposure to hypercapnic hypoxia in vivo (before cell isolation for culture) compared to hypoxia alone, suggesting a potentiating effect of hypercapnia. Similarly, analogous data were observed after exposure to hypercapnic hypoxia in vitro (post cell isolation for culture) preceding oxygen–glucose deprivation modeling. However, following exposure to hypercapnic hypoxia preceding focal cerebral ischemia modeling in rats, the content of cells expressing HIF-1α in the peri-infarct region was lower than after exposure to hypoxia alone.

It is important to note that high levels of HIF-1α can disrupt apoptotic processes in cells [111,172] and their energetic homeostasis [173], often associated with many oncogenic conditions [174]. Consequently, excessive elevation of HIF-1α levels may be considered an unfavorable factor in the pathogenesis of acute ischemic injury.

These facts suggest that the hypercapnic component, when combined with hypoxia, modulates the signaling mechanism of HIF-1 activation under conditions of damaging hypoxia/ischemia without directly affecting its accumulation. Such a mechanism may be directed towards stimulating cellular adaptive potential and increasing the tolerance of nervous tissue to ischemia/hypoxia by protecting against excessive HIF-1α accumulation in response to moderate hypoxia [175].

Modeling focal ischemic brain injury in rats confirms that hypercapnia does not affect HIF-1α under isolated conditions, but when combined with hypoxia, it reduces its content in tissues. However, in the context of a greater neuroprotective potential of the combination of hypoxia with hypercapnia, this can be considered as a protective mechanism against excessive HIF-1 activation (Figure 7).

## 6. Future Directions and Potential for Clinical Application

The phenomenon of potentiating hypoxia’s protective effects with hypercapnia is of high interest for clinical application. Particularly promising is the use of hypercapnic hypoxia in cases where the application of intermittent hypoxic training is limited due to the duration of exposure and the large number of sessions within one course (1–15 h hypoxic exposure with at least seven sessions) [28,176,177]. Additionally, intermittent hypercapnic–hypoxic exposure has been shown to achieve protective effectiveness after only three exposures, and increasing the frequency of exposures is accompanied by a proportional increase in resistance [7]. However, combining respiratory hypercapnic–hypoxic exposures with pharmacological modulators of neuro-/cardio-/cytoprotection signaling pathways appears to be the most promising therapeutic strategy. The additional efficacy when combining neuroprotectants may be moderate, additive, or even synergistic in some cases [178].

A promising approach for modulating the effects of HIF-1α during hypoxic training involves combining interventions that affect both the traditional and alternative mechanisms of its activation [179,180]. The combination of agents that enhance HIF-1α activity with respiratory interventions and drugs affecting other adaptive mechanisms may be most effective. This is supported by the increased cardioprotective effect observed when combining hypoxic preconditioning with opioid receptor antagonists and agonists [181,182], the neuroprotective effect of hypercapnic–hypoxic exposures when combined with ATP-dependent potassium channel activators and adenosine receptors [65], and angiotensin-converting enzyme inhibitor [183].

Based on data on neuroprotective mechanisms, signaling pathways can be identified, the additional stimulation of which has the potential to enhance protective effects when combined with respiratory hypercapnic–hypoxic interventions. For example, pharmaceuticals possessing antioxidant efficacy and exerting endothelial–protective effects may have potential for enhancing the effectiveness of hypercapnic hypoxia [184]. Effective potentiation may also be achieved through prolonging the effect of hypercapnia by carbonic anhydrase inhibition [185].

It can also be hypothesized that the inhibition of JNK using synthetic pharmaceuticals (e.g., IQ-1S), which exert a pronounced cardio-/neuroprotective effect, may have a high potential for enhancing the protective effects of hypercapnic hypoxia [186].

Clinical studies have demonstrated the effectiveness of hypercapnic–hypoxic training in the treatment of neurological disorders and diseases. For example, in a randomized triple-blind placebo-controlled study, respiratory training with hypercapnic hypoxia was shown to have a positive effect on the functional state of the nervous system in children with cerebral palsy and can be used as a means to increase the effectiveness of standard therapy for these patients [187]. Additionally, the effectiveness of hypercapnic hypoxia has been demonstrated in the treatment of diabetic polyneuropathy in pediatric patients [188]. Such efficacy is likely associated with the high neuroplasticity and adaptogenic potential in childhood [189].

Another promising application for hypercapnic–hypoxic breathing training is the neurorehabilitation of patients with ischemic stroke. Particularly significant are the promising results of a pilot clinical study on the effectiveness of neurorehabilitation potential of hypercapnic–hypoxic training in patients in the early period after ischemic stroke [190].

Interestingly, intermittent hypercapnic–hypoxic exposures find application not only in rehabilitative medicine and neurology but also in urological and dental practice, which can be explained by their positive influence on microcirculation parameters in organs and tissues [191]. For example, it has been noted that the application of a course of hypercapnic–hypoxic exposures in men improves perfusion of the prostate gland in chronic abacterial prostatitis [192] and microhemocirculation in the salivary glands after sialolithectomy procedures [193].

Commercial divers who engage in breath-hold diving (referred to as “Ama” in Japan) are subjected to frequent and prolonged episodes of hypercapnic hypoxia [194]. This circumstance, in turn, may decrease the likelihood of ischemic stroke and enhance the resilience of the brain to ischemic-reperfusion injury. However, these individuals are also exposed to elevated atmospheric pressure and low temperatures while underwater, and breath-hold sessions are repeated multiple times within a relatively short period. As a result, there is a high risk of disrupting adaptive mechanisms, inducing oxidative stress [195], and developing decompression illness, which leads to ischemic brain injuries [194,196]. This may also be attributed to psychosocial stress, which induces hyperventilation and hypocapnia in divers. However, it raises an intriguing possibility of the brain’s ischemic tolerance formation after hypercapnic–hypoxic exposures accompanying breath-hold episodes in commercial divers experiencing moderate stress.

## 7. Conclusions

The hypothesis formulated regarding the mechanism of mutual potentiation of neuroprotective effects of hypercapnia and hypoxia is supported by data on the molecular and cellular mechanisms during their combined intermittent exposure. Most of the molecular and cellular mechanisms involved in the formation of brain tolerance to ischemia are influenced by both excess carbon dioxide and oxygen deficiency (Figure 8). However, some signaling systems are associated with the predominance of only one of the components of hypercapnic hypoxia exposure. For example, molecules such as HIF-1α and A1 receptors are primarily influenced by hypoxic stimuli, while the NF-κB factor and maintenance of selective blood–brain barrier permeability are mainly modulated by hypercapnia.

An important fact is that carbon dioxide can have a limiting effect on the hypoxic activation of protective mechanisms such as A1 receptors and the transcription factor HIF-1α. It can be assumed that this feature allows tissues to achieve greater tolerance to ischemia/hypoxia at the organismal level and prolong adaptive changes in the long term.

## Figures and Tables

**Figure 1 ijms-25-03665-f001:**
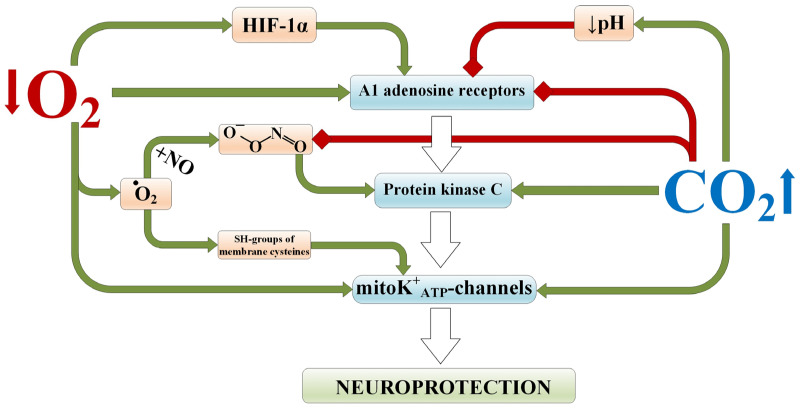
Involvement of A1 receptors and mitoK^+^_ATP_-channels in the molecular mechanisms of neuroprotective efficacy of hypercapnia and hypercapnic hypoxia [53]. Red lines indicate inhibition; green lines indicate activation/induction. HIF-1α—hypoxia-inducible factor 1-alpha; mitoK^+^_ATP_ channels—mitochondrial ATP-sensitive potassium channels.

**Figure 2 ijms-25-03665-f002:**
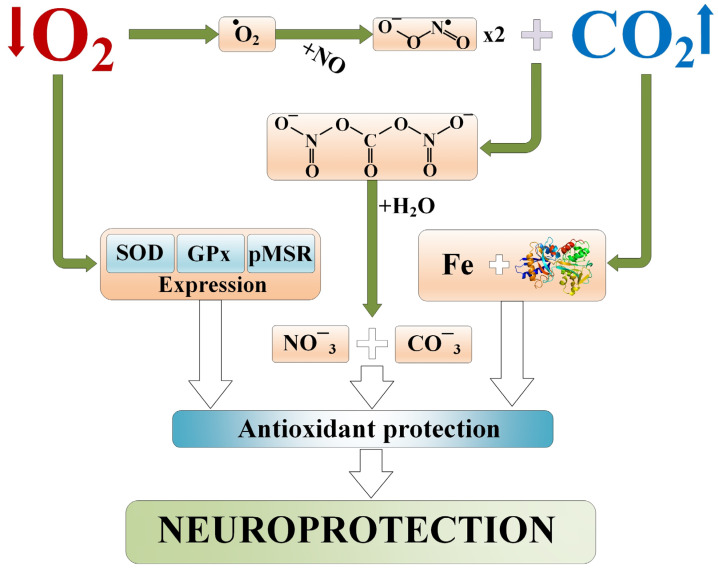
Antioxidant protection during permissive hypercapnia. SOD—Superoxide Dismutase; GPx—Glutathione Peroxidase; pMSR—Peptide Methionine(R)-S-Oxide Reductase.

**Figure 3 ijms-25-03665-f003:**
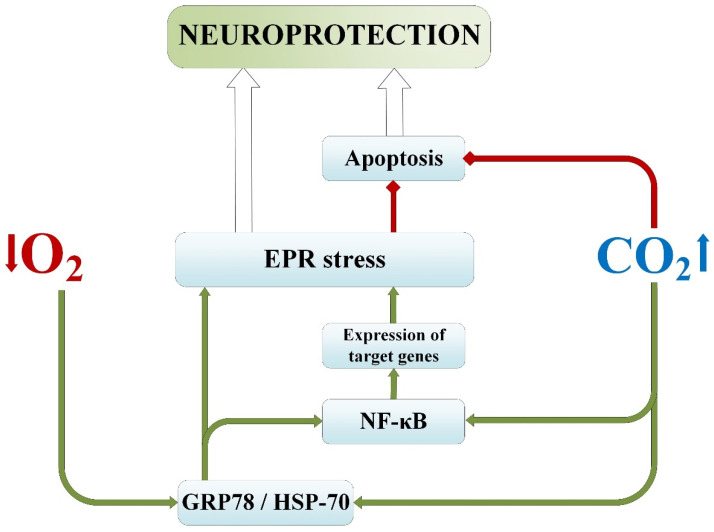
Impact of permissive hypercapnia and normobaric hypoxia on the chaperone GRP-78 and NF-κB factor. Green lines—activation/induction. Red lines—inhibition. EPR—endoplasmatic reticulum; HSP-70/GRP-78—the 70-kilodalton heat shock proteins; NF-κB—Nuclear factor kappa-light-chain-enhancer of activated B cells.

**Figure 4 ijms-25-03665-f004:**
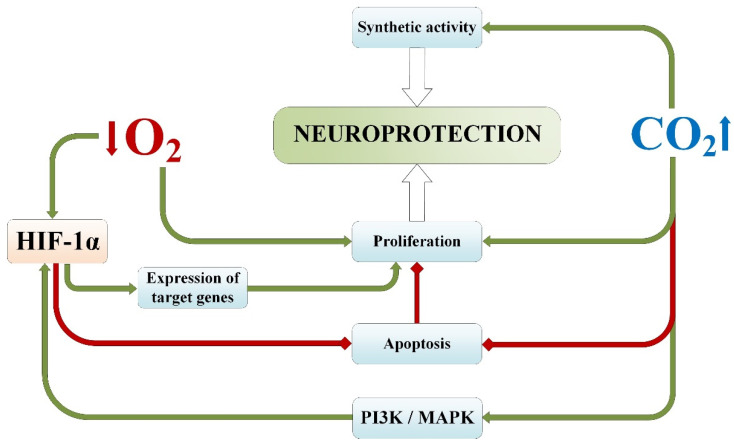
Impact of permissive hypercapnia and normobaric hypoxia on signaling pathways enhancing the synthetic and proliferative activity of nerve cells. Green lines indicate activation/induction, while red lines indicate inhibition. HIF-1α—hypoxia-inducible factor 1-alpha.

**Figure 5 ijms-25-03665-f005:**
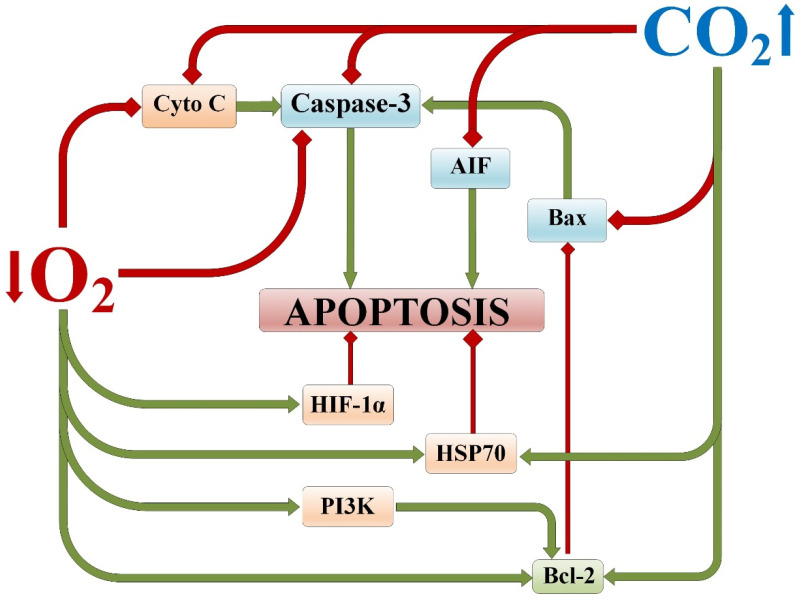
Impact of permissive hypercapnia and normobaric hypoxia on the key mediators of apoptotic signaling pathways [128]. Red lines indicate inhibition, while green lines indicate activation/induction. Bax—Bcl-2-associated X protein; Bcl-2—B-cell lymphoma 2; AIF—Apoptosis-inducing factor; HIF-1α—hypoxia-inducible factor 1-alpha; HSP-70—the 70-kilodalton heat shock proteins; PI3K—Phosphoinositide 3-kinases.

**Figure 6 ijms-25-03665-f006:**
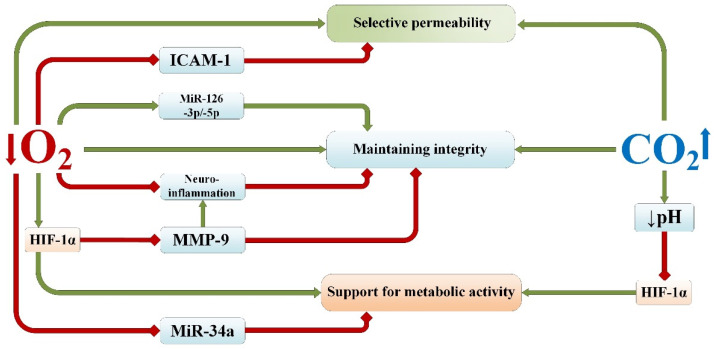
The impact of permissive hypercapnia and normobaric hypoxia on signaling pathways regulating the structural integrity of the blood–brain barrier, its selective permeability, and metabolic activity. Red lines—inhibition; green lines—activation/induction. miR—Micro ribonucleic acid; MMP-9—Matrix metalloproteinase-9; ICAM-1—Intercellular Adhesion Molecule 1; HIF-1α—hypoxia-inducible factor 1-alpha.

**Figure 7 ijms-25-03665-f007:**
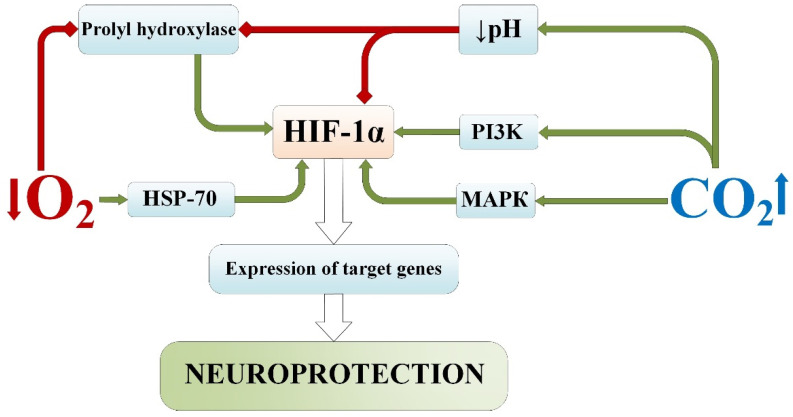
Signaling pathways of the influence of permissive hypercapnia and normobaric hypoxia on the transcription factor HIF-1α. Red lines—inhibition; green lines—activation/induction. MAPK—Mitogen-activated protein Kinase; HIF-1α—hypoxia-inducible factor 1-alpha; HSP-70—the 70-kilodalton heat shock proteins; PI3K—Phosphoinositide 3-kinases.

**Figure 8 ijms-25-03665-f008:**
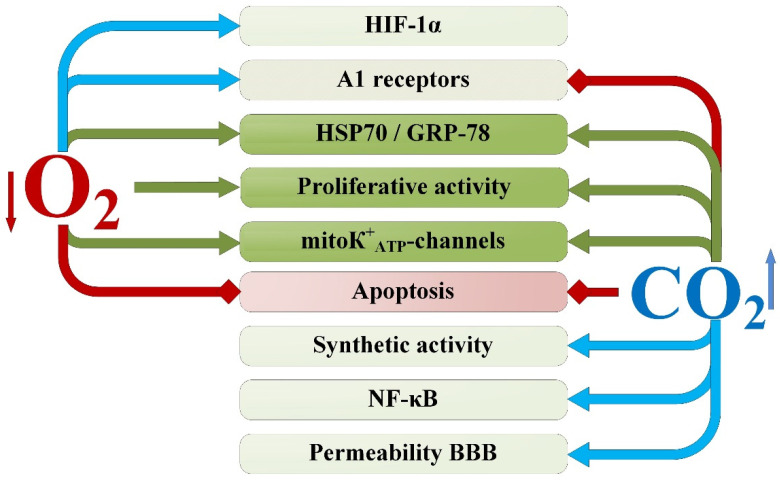
Potentiation of neuroprotective mechanisms during exposure to hypercapnia and hypoxia. Arrows indicate the directions of the detected effects of excess CO_2_ and oxygen deficiency: red lines—inhibition; green lines—activation/induction by both factors; blue lines—activation/induction by only one factor. HSP-70/GRP-78—the 70-kilodalton heat shock proteins; HIF-1α—hypoxia-inducible factor 1-alpha; BBB—blood–brain barrier; mitoK^+^_ATP_ channels—Mitochondrial ATP-sensitive potassium channels; NF-κB—Nuclear factor kappa-light-chain-enhancer of activated B cells.

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
