# Peer review of "Molecular Mechanisms of Neuroprotection after the Intermittent Exposures of Hypercapnic Hypoxia"

_ijms, 2024, doi:10.3390/ijms25073665_

Round 1
Reviewer 1 Report
Comments and Suggestions for Authors
The manuscript Molecular mechanisms of neuroprotection after the intermittent exposures of hypercapnic hypoxia covers a rather interesting topic related to hypercapnic-hypoxic training and its neuroprotective efficacy as well as the underlying molecular mechanisms. The manuscript is well organized, presents the topic clearly and is scientifically attractive. I firmly believe that the manuscript will be of great interest to the readers of the journal IJMS after a minor revision, i.e., after taking into account the suggestions and corrections listed below.
1. Since the scientific core of the review is hypercapnic-hypoxic training and its neuroprotective potential, please define the term permissive hypercapnia as well as hypoxia more precisely. For example, what is the level and duration of hypercapnia to be considered permissive and neuroprotective, etc.?
2. As for the information provided in Subsection Maintenance of Electrolyte Balance and Ca2+ Homeostasis, the data mainly focus on the “ischemia-resistant neurons” of the CA1 hippocampus. What about other brain regions and neurons? Please clarify this aspect.
3. I would suggest renaming the Subsection 2.3. Intracellular Signaling Remodeling into for example Intracellular Signaling Remodeling associated with adenosine and its receptors.
4. While reading the manuscript, I was really intrigued by a potentially interesting connection between professional divers and hypercapnic-hypoxic training which is not discussed in the manuscript. Given the nature of their profession, these individuals are exposed to frequent hypercapnic-hypoxic training. Is there any data in the literature on commercial divers and possible neuroprotection in diseases such as ischemic stroke or neurodegenerative disorders? There is a possibility that these individuals may be less likely to suffer an ischemic stroke, for example, or at least be less susceptible to damage after an ischemia/reperfusion event. I propose to address this topic in Section 6, Future directions and potential for clinical application.
Author Response
The authorial team sincerely thanks you for the careful review of our manuscript and expert assistance! We hope that we have been able to address your queries, and our manuscript will have a better appearance and quality after adjustments according to the provided recommendations.
Below are the responses to your questions and recommendations, which have also been incorporated into the manuscript:
- Corrections have been made to the manuscript (Section "1" - paragraphs 1 and 2).
- Indeed, information about the mechanisms of neuron resilience to ischemia/hypoxia mentioned in subsection 2.1 often pertains specifically to neurons from the CA1 region in the hippocampus. This is because this region exhibits the highest sensitivity to hypoxia, and therefore, the increase in neuron resilience to ischemia/hypoxia in this area of the brain is considered a characteristic feature of the formation of ischemic tolerance in the brain [Obrenovich T., 2008]. Thus, many experimental studies regarding the mechanisms of ischemic tolerance formation have been conducted for the CA1 region of the hippocampus and can be extrapolated to other areas of the brain. Corrections have been made to the manuscript (Section "2.1" - paragraph 5).
- Corrections have been made to the subsection title 2.3 "Remodeling of intracellular signaling associated with adenosine, its receptors, and mitoK+ ATP channels".
- First and foremost, we would like to express our gratitude for your valuable recommendation regarding further clinical research! Indeed, commercial divers who engage in breath-hold diving (referred to as "Ama" in Japan) are subjected to frequent and prolonged episodes of hypercapnic hypoxia. This circumstance, in turn, may decrease the likelihood of ischemic stroke and enhance the resilience of the brain to ischemic-reperfusion injury. However, these individuals are also exposed to elevated atmospheric pressure and low temperatures while underwater, and breath-hold sessions are repeated multiple times within a relatively short period. As a result, there is a high risk of disrupting adaptive mechanisms, inducing oxidative stress [Salah H, El-Gazzar RM, Abd El-Wahab EW, Charl F. Oxidative stress and adverse cardiovascular effects among professional divers in Egypt. J Occup Environ Hyg. 2023 Mar-Apr;20(3-4):159-169. doi: 10.1080/15459624.2023.2173364], and developing decompression illness, which leads to ischemic brain injuries [Kohshi K, Denoble PJ, Tamaki H, Morimatsu Y, Ishitake T, Lemaître F. Decompression Illness in Repetitive Breath-Hold Diving: Why Ischemic Lesions Involve the Brain? Front Physiol. 2021 Sep 3;12:711850. doi: 10.3389/fphys.2021.711850]. This may also be attributed to psychosocial stress, which induces hyperventilation and hypocapnia in divers. However, it raises an intriguing possibility of the brain's ischemic tolerance formation after hypercapnic-hypoxic exposures accompanying breath-hold episodes in commercial divers experiencing moderate stress. Corrections have been made to the manuscript (Section "6" - paragraph 8).
Reviewer 2 Report
Comments and Suggestions for Authors
The authors of the study titled „Molecular mechanisms of neuroprotection after the intermittent exposures of hypercapnic hypoxia“ discusses the role of molecular-cellular signaling pathways in the mechanism of neuroprotection during intermittent exposure to hypercapnia and/or hypoxia preceding ischemic damage.
A. The proficiency in the English language in this study is satisfactory.
B. The title effectively captures the scope of the study.
C. Abstract: The abstract is not that well written. It lacks information and requires improvement. While it's important to cover various molecular-cellular signaling pathways and mechanisms, the abstract could benefit from being more concise and focused. Try to streamline the content to include only the most relevant information. Some sentences are lengthy. Additionally, ensure smoother transitions between different topics. Consider adding a concluding sentence to summarize the key implications or findings of the review.
D. Introduction: The introduction lacks clarification for terms such as hypoxia, hypercapnia, and ischemia...they should be briefly explained.
The text briefly mentions the involvement of signaling molecules like HIF-1α and protein kinase C but how these signaling cascades interact with adenosine receptors and mitoK+ATP channels to modulate neuroprotection in response to hypoxia and hypercapnia?
Focal ischemic stroke should be briefly explained.
Line 222: Address why peroxynitrite is considered harmful.
The text introduces the roles of heat shock proteins (HSP-70) and the chaperone GRP-78 in cellular stress response but lacks an explanation of their functions.
Line 363: Does all types of brain ischemia lead to the formation of penumbra?
Line 368: Why is apoptosis preferable to necrosis? Could you briefly explain the concepts of apoptosis and necrosis in the manuscript? This would provide clarity and make it easier to understand why apoptosis is preferable to necrosis, it would be easier to understand the statement: „Additionally, in the context of therapeutic intervention, apoptosis is preferable to necrosis, as it can be blocked by various treatment methods, allowing the preservation of partially damaged tissue [116,119].“.
Line 379:„ p53-dependent mechanisms“- please briefly explain in the manuscript.
Overall, the paper " Molecular mechanisms of neuroprotection after the intermittent exposures of hypercapnic hypoxia " is well-written, but it would benefit from a deeper interpretation of the literature findings, rather than simply listing them in the text. Given the appealing subject matter, it is likely to attract a broad readership. therefore, the paper can be considered for publication after correction.
Comments on the Quality of English LanguageMinor errors detected.
Author Response
We thank you for your objective assessment of our work and the questions formulated regarding the manuscript text!
Below are the responses to your questions and recommendations, which have also been incorporated into the manuscript:
C. Abstract: The abstract is not that well written. It lacks information and requires improvement. While it's important to cover various molecular-cellular signaling pathways and mechanisms, the abstract could benefit from being more concise and focused. Try to streamline the content to include only the most relevant information. Some sentences are lengthy. Additionally, ensure smoother transitions between different topics. Consider adding a concluding sentence to summarize the key implications or findings of the review.
- Corrections have been made to the manuscript (Abstract section).
D. Introduction: The introduction lacks clarification for terms such as hypoxia, hypercapnia, and ischemia...they should be briefly explained.
- Corrections have been made to the manuscript (Section "1" - paragraphs 1 and 2).
The text briefly mentions the involvement of signaling molecules like HIF-1α and protein kinase C but how these signaling cascades interact with adenosine receptors and mitoK+ATP channels to modulate neuroprotection in response to hypoxia and hypercapnia?
- Hypoxia induces accumulation of HIF-1α factor, which enhances the activity of adenosine receptors [according to Howell N.J. and Tennant D., 2014]. A1 receptors activate protein kinase C [Martire A. et al., 2019], which leads to modification of mitoK+ATP channels [Ilie A. et al., 2009].
- The formation of peroxynitrite under hypoxic conditions also leads to subsequent activation of protein kinase C [according to Krenz M. et al., 2002]. Activation of this kinase also occurs with an increase in CO2/bicarbonate concentration and intracellular Ca2+ elevation [according to Obrenovitch T.P., 2008].
- We have indicated in Figure 1 a schematic interaction of these signaling molecules with each other, combining the results of our own research and data obtained by other authors. Additionally, we have made corrections to the manuscript (Section "2.3" - paragraph 1), supplementing it with missing information on the interaction between protein kinase C, adenosine receptors, and mitoK+ATP channels.
Focal ischemic stroke should be briefly explained.
- Corrections have been made to the manuscript (Section "2.3" - paragraph 8). Focal ischemic stroke is modeled in rodents by focusing light or laser beams in a specific location of the brain (most commonly the cortex) followed by artery thrombosis. The key feature of this model is the focal localization with clear boundaries of the ischemic damage zone. Moreover, such a model exhibits characteristics of the most common pathogenetic variants of ischemic stroke in humans – atherothrombotic and cardioembolic ischemic stroke.
Line 222: Address why peroxynitrite is considered harmful.
- Corrections have been made to the manuscript (Section "2.4" - paragraph 1).
The text introduces the roles of heat shock proteins (HSP-70) and the chaperone GRP-78 in cellular stress response but lacks an explanation of their functions.
- Corrections have been made to the manuscript (Section "2.5" - paragraph 1).
Line 363: Does all types of brain ischemia lead to the formation of penumbra?
- Certainly, not all types of cerebral ischemia lead to the formation of the penumbral zone, but only those that create a restriction of blood flow without its complete cessation in the region surrounding the core of the stroke. We have made corrections to the manuscript (section "3.2" - paragraph 1), replacing the term "penumbral zone" with "peri-infarct zone." This term is widely used in studies focused on investigating apoptotic signaling pathways in the mechanism of neuroprotection and the formation of ischemic tolerance in the brain.
Line 368: Why is apoptosis preferable to necrosis? Could you briefly explain the concepts of apoptosis and necrosis in the manuscript? This would provide clarity and make it easier to understand why apoptosis is preferable to necrosis, it would be easier to understand the statement: „Additionally, in the context of therapeutic intervention, apoptosis is preferable to necrosis, as it can be blocked by various treatment methods, allowing the preservation of partially damaged tissue [116,119].“.
- Corrections have been made to the manuscript (section "3.2" - paragraph 1).
Line 379:„ p53-dependent mechanisms“- please briefly explain in the manuscript.
- Corrections have been made to the manuscript (section "3.2" - paragraph 2).
Reviewer 3 Report
Comments and Suggestions for Authors
The review summarizes the current knowledge about hypercapnia/hypoxia on neuroprotection. Sections are well written, the introduction gives a general overview about the history of permissive hypercapnia research and the importance of gaining deeper knowledge about the phenomenon.
Section 2, 3, 3 and 5 sums up those cellular and molecular mechanisms that are involved in ischaemia-induced tolerance, however, and extra section about microglial mechanism would improve the quality of the manuscript and would support to learn the immunological aspects of the preventive mechanisms especially in case of NF-κB and HIF-1α. Potential clinical applications are summarized at Section 6 and Section 7 describes the hypothesis of the authors about the neuroprotective mechanisms of hypercapnia and hypoxia.
The strong points of the manuscript are the summary figures at the end of each sections and the well written descriptive parts.
English language is fine, no issues were detected.
The manuscript is acceptable in the present form.
Author Response
Dear Reviewer,
We sincerely appreciate your thorough review our manuscript and for the positive evaluation of our work. Your recommendation regarding the additional section on microglial mechanisms of neuroprotective efficacy is highly valuable. Currently, we are conducting new experimental research aimed at investigating the signaling pathways and cellular mechanisms of neuron-glia interaction following exposure to hypercapnia and/or hypoxia. In the publication of the results of this study, we will definitely include a section containing information on immunological aspects and modulation of neuroinflammation in the context of neuroprotection.
Round 2
Reviewer 2 Report
Comments and Suggestions for Authors
The authors answered all my questions and made the corrections that I suggested. It is easier to follow the text now.
Comments on the Quality of English LanguageMinor errors detected.